# Out-of-Distribution Generalization in the ARC-AGI Domain: Comparing Execution-Guided Neural Program Synthesis and Test-Time Fine-Tuning

## Abstract

We run a controlled compositional generalization experiment in the ARC-AGI domain: an open-world problem domain in which the ability to generalize out-of-distribution is, by design, an essential characteristic for success. We compare neural program synthesis and test-time fine-tuning approaches on this experiment. We find that execution-guided neural program synthesis outperforms all reference algorithms in its ability to compose novel solutions. Our empirical findings also suggest that the success of TTFT on ARC-AGI lies mainly in eliciting in-distribution knowledge that the LLM otherwise fails to rely on directly.

## 1 Introduction

ARC-AGI and, more recently, ARC-AGI-2 (Chollet et al. (2025)), is a challenging open-world visual reasoning problem domain. At its core, ARC-AGI assesses whether a system can recombine primitive operations, or prior knowledge, in novel ways to solve unseen problems; a notion known as compositional generalization. Human solvers effortlessly apply compositional reasoning, but achieving this efficiently in machines has remained elusive. The ARC-AGI hidden test set enforces strict separation between training and final evaluation. While participants can refine models on the public tasks, final performance is reported on tasks with no prior examples online, guarding against overfitting and rewarding genuine generalization.

While ARC-AGI may not capture all aspects of AGI, it offers a concrete test of a critical capability: solving structurally novel reasoning tasks by recombining known primitives. This could be one of the main bottlenecks preventing us from reaching AGI in any deep or strong sense of the word. Indeed, cutting edge AI systems fail to achieve results above 29% on ARC-AGI-2 at the time of this writing. The fact that humans solve these problems while our best AI systems do not is direct, practical evidence of a fundamental missing functionality with respect to AGI.

We are interested in better understanding the performance of neural program synthesis methods, especially execution-guided, multi-step neural program synthesis, in comparison to test-time fine-tuning approaches. Since execution-guided neural program synthesis has not yet been attempted on ARC-AGI, we implement a version of it in this domain, inspired by the most recent literature.

In summary, our main contributions are:

- We present an implementation of an execution-guided multi-step neural program synthesis (EG-NPS) algorithm for ARC-AGI, along with a new DSL and program syntax that facilitates the latter.

- We present a carefully controlled out-of-distribution generalization experiment to compare EG-NPS, non-execution-guided NPS, and test-time fine-tuning (TTFT).

## 2 RELATED WORK

**Neural Program Synthesis on ARC-AGI** On the 2024 ARC-AGI competition, two papers described a NPS approach: the Latent Program Network (Bonnet & Macfarlane (2024)) and GridCoder (Ouellette (2024)). The latter is a non-execution-guided approach based on a CNN+Transformer-architecture and a relatively high-level, specific DSL. The former differs from more traditional program synthesis in that no hand-made DSL is required at all; the algorithm implicitly learns the required primitives from the training data. On the public score board, there is a notable, high-performing NPS algorithm that was attempted (Greenblatt (2024)). It uses an LLM, not even fine-tuned on ARC-AGI, to generate program candidates and refine them via trial and error.

**Execution-guided Program Synthesis** Zohar & Wolf (2018) introduce PCCoder: at each synthesis step, a model predicts the next high-level operation, its operands, and which variables can be dropped ("garbage-collected"), based on the current program state and I/O context. The garbage collection is learned by a distinct neural network than the synthesis. They demonstrate that they are able to create programs that are more than twice as long as existing solutions, while improving the success rate for comparable lengths, and cutting the run-time by two orders of magnitude. Ellis et al. (2019) build on the latter, but use a reinforcement learning approach for the program search instead of a beam search.

Shi et al. (2024) enhance the approach by having a separate Transformer model, the subgoal network, that predicts the next goal state to reach. Doing this decomposes the larger problem in sub-problems which further improves performance on tasks. Verbruggen et al. (2025) introduce execution-guided within-prompt search, a method that iteratively generates and evaluates candidate lines of code within a single prompt. Execution feedback on each candidate is incorporated directly into the prompt as annotations, allowing the language model to refine its output in subsequent generations.

Lavon et al. (2025) propose Execution-Guided Classifier-Free Guidance, a line-by-line LLM-based code generation method that integrates real-time execution feedback into the decoding process. At each line, the model samples multiple candidate continuations via beam search, executes them against test cases, and constructs an execution-trace prompt. Classifier-Free Guidance then steers the next tokens by interpolating between the normal and execution-conditioned distributions, refreshing this feedback at each line boundary.

**Test-time fine-tuning** Akyürek et al. (2025) use an `8B Llama 3` model pretrained on synthetically generated tasks, and they evaluate on a subset of the publicly available ARC-AGI validation tasks. They show an increase in performance, as in the empirical results we will present, but contrary to their claims, it cannot be concluded that this is due to an ability to generalize to structurally novel tasks: they use a pretrained foundation model and evaluate on publicly available data. In addition, they generate tasks synthetically for training, as well as geometric transformations thereof, that can overlap with validation data. Thus, data contamination is likely, and whether their evaluation benchmarks are truly out-of-distribution is unknown.

## 3 BACKGROUND

ARC-AGI consists of solving tasks that are composed of (typically 3 to 6) pairs of grids. Each pair consists of an input grid and its associated target grid. Each grid can have a size of anywhere between $1 \times 1$ to $30 \times 30$ cells, each cell being one of 10 possible colors. Each task has its own logic that converts the input grid examples to their associated target grids. The solver must figure out the underlying logic or algorithm, and apply it to a test grid in order to confirm its understanding. It is evaluated on a hidden private test set, which consists of tasks that are meant to be distinct from any of the tasks that are publicly available.

Additionally, in ARC-AGI-2, submission attempts are not immediately scored on this private test set, to avoid "data mining" or "overfitting" this test set. This methodology encourages building solvers that can adapt to novelty, instead of simply interpolating over a predefined domain.

## 4 EXECUTION-GUIDED SYNTHESIS FOR ARC-AGI

We implement a variant of execution-guided neural program synthesis for ARC-AGI. This implies developing our own DSL for this domain, in a way that facilitates having a tractable, tokenizable intermediate state at each step along the way.

### 4.1 EG-NPS

The execution-guided neural program synthesis implementation we propose here uses a DSL and a Transformer trained from scratch, rather than an LLM with pretraining. Unlike Zohar & Wolf (2018), our memory management strategy is based around the `del` primitive. Whenever this instruction occurs, the specified state variable is removed from memory. Thus, we have one neural network, not two, and its responsibility is both to help synthesize the program and to manage memory in the same process.

The execution-guided feedback is implemented by executing the current instruction step, tokenizing its output (in our DSL, each instruction step always outputs exactly one variable), feeding it to the encoder to produce an embedding, and concatenating this embedding to the previous ones. This generated encoder memory is used as input to the decoder for the next instruction step decoding iteration (see Figure 1).

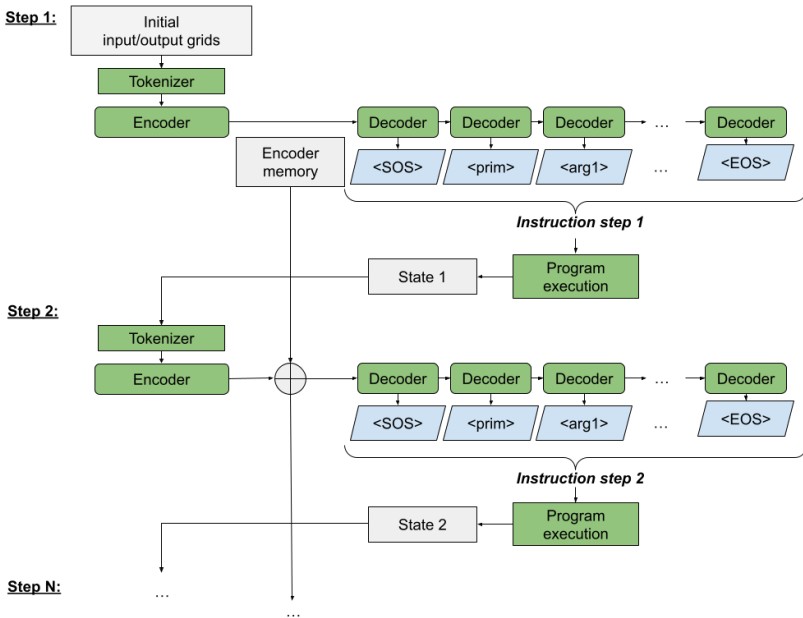

Figure 1: A diagram of the state-conditioned, execution-guided inference used. Each step consists of decoding a single instruction sequence from looking only at a concatenated accumulation of past encoded states and the encoded target state. The resulting token sequence is executed, the output is tokenized and encoded, added to encoder memory, and the procedure is repeated.

Additionally, a feature we refer to as "entropy" was added to take up the remaining time budget when the full probability space is explored well before its expiration. When that happens, it re-launches the search, but in this second iteration it adds a fixed probability value to randomly selected tokens of the root node expansion. This creates further exploration of paths that previously had a zero probability assigned to them, potentially discovering new trajectories that solve the problem. Although the impact of this mechanism on aggregate success rates was limited, it enabled the solution of three additional task instances that could not be solved without it.

Our program search algorithm is a tree search in which each node represents the application of one particular instruction step sequence. Each node that gets expanded is populated with a list of non-zero probability instruction sequences conditional on that node's program state. Each child of this

node contains an index indicating which of these instruction sequences was executed to reach that child node. Thus, each node also contains a state variable, corresponding to the output of applying its instruction step sequence.

The search through this tree continues until either a solution is found or a timeout occurs. A solution is considered found when the generated program state is exactly like the expected target grids of the demonstration set examples. Details of this algorithm can be found in Appendix A, but the high-level iterative steps are:

1. Collect the currently selected node's input state by concatenating the intermediate state of all its parents in the tree. Execute the selected node's instruction step on this state. Save the result inside that node.

2. If this result is the target grid set, a solution has been found.

3. Given the selected node, the target grids, and the trained model: generate all token sequences for the next instruction step (whose probability > some threshold, 0.01 in our case).

4. Calculate the joint probability of each of these instruction step token sequences.

5. Insert these new programs into a global queue, ordered by joint probability. The new programs are the programs represented by the tree path down to the selected node, to which each of the newly generated instruction steps are appended.

6. This program queue is used to select the next program to attempt (the highest probability one), which is then dequeued. A new child node is instantiated from this next program's last instruction step. This becomes the new selected node. Repeat from step 1.

The neural network used is a standard encoder-decoder transformer, whose hyperparameters are detailed in Appendix B. During training, each ground truth program is decomposed into its individual instruction steps. Each training sample is a token sequence of an instruction step, coupled with the tokenized sequences of intermediate states that have been generated in the previous instruction steps. This includes the initial input-output grid set.

This tokenized intermediate state is fed to the encoder, which produces the embedding that is fed to the decoder. The latter predicts a token sequence that minimizes the cross-entropy loss with respect to the ground truth instruction step. This supervised training proceeds iteratively: each iteration, the encoder embedding for subsequent steps is generated by executing the instruction steps from previous iterations and concatenating them to the current embedding.

## 5 DSL

We introduce a custom domain-specific language (DSL) tailored for ARC-AGI. Each program is a sequence of instruction steps, where each step executes a single primitive operation and produces exactly one output variable (e.g., a Grid, a list, or a scalar). Each instruction is formatted as a token sequence (see Table 1)

```
[<primitive>, <SEP>, <arg1>, <ARGSEP>, <arg2>,
<ARGSEP>, ..., <argN>, <EOS>]
```

| Component | Description |
|---|---|
| `<primitive>` | Operation to apply (e.g., `equal`, `set_pixels`) |
| `<SEP>` | Separator between primitive and arguments |
| `<argN>` | Argument: constant, object attribute, or reference to prior variable |
| `<ARGSEP>` | Separator between arguments |
| `<EOS>` | Marks the end of the instruction step |

Table 1: DSL instruction structure. Each instruction is a token sequence of the form `[<primitive>, <SEP>, <arg1>, <ARGSEP>, ..., <argN>, <EOS>]`.

Each argument can be a constant or a reference to an existing variable in the current state of the program, either the initial input grid set or the output of a previous instruction step. Accessing a Grid's attributes is done by using two consecutive argument tokens: the reference to the object, and the attribute token. There is a special `del` primitive that does garbage collection of specified program state variables that are no longer needed by the rest of the program. Table 2 describes the token types and their range. See Appendix D for details.

| Token Range | Meaning |
|---|---|
| `0--9` | Integer constants |
| `10--N` | Primitives and object attribute tokens (e.g., `.x`, `.c`) |
| `N+0` to `N+max` | State variable references to intermediate values (e.g., outputs of previous instruction steps) |

Table 2: Token ranges in the DSL vocabulary. N is the total number of primitives (including the constants and object attributes) in the DSL. The vocabulary includes integer constants, DSL primitives, object attributes, and references to previously defined state variables.

The example program in Table 3 contains four instruction steps. Its effect is to change all non-zero pixels of the grid to the color 2. The first instruction returns a boolean list for each pixel in the input grid set (reference: $N + 0$), set to true if 0, false otherwise. Instruction 2 is an "if/else" statement going over each element of $N + 1$. When its value is True, it outputs 0, otherwise it outputs 2. The third instruction removes from memory the now obsolete $N + 1$. The last instruction sets all pixels of the input grid set $(N + 0)$, at positions determined by .x and .y attributes of $N + 0$ (i.e., all pixels are modified), to the color value specified in what was previously $N + 2$, but is now $N + 1$ due to the previous `del` operation.

#1.   `[equal, <SEP>, N+0, <.c>, <ARGSEP>, 0, <EOS>]`    $\rightarrow N + 1$
#2.   `[switch, <SEP>, N+1, <ARGSEP>, 0, <ARGSEP>, 2, <EOS>]`    $\rightarrow N + 2$
#3.   `[del, <SEP>, N+1, <EOS>]`
#4.   `[set_pixels, <SEP>, N+0, <ARGSEP>, N+0, <.x>,`
      `<ARGSEP>, N+0, <.y>, <ARGSEP>, N+1, <EOS>]`

Table 3: Example DSL program: recolors all non-zero pixels in the input grid to color 2.

## 6   Experiments & Results

The controlled experiment presented here is designed to clearly compare the capabilities of competing approaches. Specifically, it examines the ability of an algorithm to learn from training data in a way that generalizes to new compositions of this training data. By knowing exactly the content of the generated training and test data, we prevent any form of data contamination. We can also evaluate in controlled ways the failure and success cases of the competing approaches.

### 6.1   The algorithms being compared

Five different approaches to solving these ARC-AGI-based problems are compared. First, the Grid-Coder implementation presented in Ouellette (2024) is evaluated. This is a neurally guided search over programs, but it does not perform state-conditioned synthesis: instead, it processes the input-output pairs and searches for the most probable program without using any feedback on the intermediate effects of its instructions. Additionally, it uses the rather specific and high-level DSL presented in that paper, rather than the lower-level, more flexible DSL used here.

`EG-NPS` is the execution-guided neural program synthesis baseline, detailed in previous sections. In `NN-Only`, the same neural network and state-conditioned inference procedure as EG-NPS is used, but without the search algorithm. Only the highest probability tokens are generated at each step, instead of searching over the space of programs for a fit on the input-output examples.

Given TTFT's popularity in ARC-AGI, we include it as a baseline. Specifically, we use the Omni-ARC algorithm (Barbadillo (2024)), which won second place on the ARC-AGI 2024 competition.

Because the comparisons must be on an equal footing, therefore trained on the same data, the LLM `Qwen2-0.5B-Instruct` is trained from scratch (i.e. without loading its pretraining weights) on the same training data as the other approaches. Additionally, because the objective of the experiments will be to evaluate out-of-distribution generalization specifically, some geometric data augmentations that can generate training data identical to the out-of-distribution tasks were removed.

Omni-ARC is a transductive approach, rather than a program synthesis approach. In other words, the inference output is not a program to execute, but rather the predicted output grid itself. It operates in essentially 4 steps:

1. The LLM (`Qwen2-0.5B-Instruct`) is fine-tuned on a training dataset designed to learn the ARC-AGI domain itself.

2. The fine-tuned LLM is then further fine-tuned on a given test task. This is the test-time fine-tuning step, which outputs a task-specific LoRA adapter.

3. The fine-tuned LLM and its associated LoRA adapter are used to perform a number of different inference attempts on the test grid(s) of the task to solve.

4. A voting procedure is used over the inference attempts to determine the true guess.

Comparison to a baseline brute-force search was omitted for the simple reason that the EG-NPS DSL search space is too vast to be tractable to such an approach under the given resource budget. All models are trained on, and experiments are run on, an A40 RunPod instance with 48Gb of VRAM, 50Gb of RAM, and 9 vCPUs (Intel(R) Xeon(R) Gold 6342 CPU @ 2.80GHz), using a three minute budget per task instance.

The training data for each task is generated by randomly sampling subgrids of ARC-AGI training tasks, and by applying the task's ground truth program to the inputs in order to obtain the target grids. The program synthesis models are trained on $200,000$ task examples each containing exactly one input/output pair, each example designed to be non-trivial and not underdetermined. The task is considered solved if it finds a program that generates the outputs (including a test output) from the input. Five runs per task example were done to evaluate variance in results (min/max/median). The DSL for the experiments #1 and #2 contain $34$ primitives, while the experiment #3 used a DSL of $45$ primitives.

The TTFT model was pretrained on a larger amount of pair examples: $100,000$ task instances each having 8 input/output demonstration pairs, for a total of $800,000$ distinct pair examples. TTFT itself is done on task instances with 3 pairs. The voting procedure, when evaluating the test output, is done over 8 different inference attempts. The most frequent output is kept.

The comparison between TTFT and EG-NPS is fair because they both have 3 minutes of test-time search (or gradient descent in the case of TTFT). If anything, the experiment is unfair to EG-NPS because it is trained on only $200,000$ whereas TTFT is trained on $800,000$ tasks. The 3 minutes time budget was selected to match the average task solution time allowed per task on the ARC-AGI Kaggle competition, and is therefore not arbitrary, nor was it selected to favor EG-NPS. `Qwen2-0.5B-Instruct` is the LLM used by Omni-ARC to reach its performance in the 2024 competition, so it has more than enough capacity to learn what it needs to learn for these experiments.

## 6.2 EXPERIMENT #1: OUT-OF-DISTRIBUTION TASKS

In this first experiment, each neural network is trained on $14$ distinct tasks that are constructed by composing a known set of high-level operations such as recoloring non-zero pixels, horizontal grid flipping, translation of pixels, etc. They are then evaluated on a different set of 7 tasks, each compositionally constructed from these same atomic operations (see Table 4).

However, these 7 test tasks are out-of-distribution (OOD) with respect to the training set because they are composed in ways that are not seen in the $14$ training tasks (see Appendix C for details). In some cases, the ground truth programs of the OOD tasks are significantly longer than the longest one seen in the training set.

For each of these OOD tasks, 10 task instances are attempted. By task instance, we mean a distinct grid input-output set that implements the same underlying task pattern. This gives a total 70 test task samples.

| OOD Task # | NN-Only | GridCoder | EG-NPS | TTFT |
|---|---|---|---|---|
| 1 | 10% | 70% | **100%** | 10% |
| 2 | 0% | **90%** | 80% | 10% |
| 3 | 10% | 70% | **100%** | 40% |
| 4 | 40% | 60% | **100%** | 0% |
| 5 | 0% | 10% | **70%** | 0% |
| 6 | 0% | 0% | **60%** | 10% |
| 7 | 10% | 0% | 50% | 0% |
| **Min** | 10% | 40.06% | **78.4%** | 9.8% |
| **Max** | 10% | 43% | **82.8%** | 12.3% |
| **Median** | 10% | 42.86% | **80%** | 10% |

Table 4: Median success rate over 10 samples for each OOD task and each algorithm, with a 3-minute budget per task sample. Total min, max and median over 5 distinct runs is shown as well.

## 6.3 EXPERIMENT #2: ANALYZING TTFT

The relative performance of TTFT in the first set of experiments can seem contradictory to the high performance of TTFT on the 2024 competition. The experiment in this section focuses entirely on TTFT, in order to better understand the sources of success, and the limitations, of this approach. In particular, what component of the performance comes from the LLM pretraining, from the various data augmentations that the original implementations use, and TTFT itself (or an interaction of these components).

This experiment is performed on the same data as Experiment #1: we fine-tune on the 14 training tasks and evaluate on the the 7 OOD tasks. In Table 5, LLM+TTFT refers to the original, intact version of the Omni-ARC implementation: the pretrained `Qwen2-0.5B-Instruct` weights are used, instead of being discarded, which means that all of the knowledge it benefits from is used. Additionally, all of the data augmentations are used, regardless of the fact that they convert tasks that were intended to be out-of-distribution into training distribution tasks.

`LLM-no-TTFT` is the full LLM, fine-tuned on the 14 training tasks, with data augmentations, but no TTFT is done. `TTFT+augments` refers to not using the `Qwen2-0.5B-Instruct` weights at all, but all of the data augmentations are kept, and TTFT is used. In this case, the LLM is fully pretrained on our custom training data instead. And finally, `TTFT-no-augments` is simply a repetition of the scores of the TTFT implementation used in Experiment #1, for visibility.

| OOD Task # | LLM+TTFT | LLM-no-TTFT | TTFT+augments | TTFT-no-augments |
|---|---|---|---|---|
| 1 | **90%** | 50% | 40% | 10% |
| 2 | **100%** | **100%** | 70% | 10% |
| 3 | **100%** | 80 % | 40% | 40% |
| 4 | **70%** | 20% | 0% | 0% |
| 5 | **80%** | 10% | 0% | 0% |
| 6 | **90%** | 0% | 10% | 10% |
| 7 | **100%** | 90% | 90% | 0% |
| **Total** | **90%** | 50% | 35.71% | 10% |

Table 5: Comparison of TTFT variants on the OOD tasks.

## 6.4 EXPERIMENT #3: ARC-AGI TRAINING TASKS

To demonstrate that our approach is not contrived and can apply to real ARC-AGI tasks, we present results on a subset of ARC-AGI-2 training tasks. To expand the model's scope, 11 new primitives and 6 additional training tasks were added. Only up to 18 tasks are theoretically solvable under the current DSL, in large part because it does not yet include the ability to detect and manipulate objects.

`EG-NPS` solves 83.33% of the theoretically solvable ARC-AGI-2 training tasks, which is 1.5% of the whole set. In contrast, `NN-Only` solved only 27.78% of the theoretically solvable tasks.

## 7 DISCUSSION

### 7.1 TTFT RESULTS

The TTFT experiments clarify why the method performs well on ARC-AGI. First and foremost, the large gap between `LLM+TTFT` and the other variants that do not use the LLM's pretraining indicates that most of the performance comes from the LLM's foundational knowledge. It is conceivable that it has already been trained on tasks that are sufficiently similar to relatively simple operations found in our OOD tasks: translating elements in a matrix, systematically replacing some values with other values, flipping the matrix elements around an axis, etc.

Yet, the performance of this same, fully pretrained LLM drops significantly when no TTFT is used (`LLM-no-TTFT`). This suggests that TTFT is able to extract information from this pretraining that is not available from direct, static inference. This is coherent with Sun & Dredze (2025), in which the authors fine-tuned multiple checkpoints of a large LLM and found that additional pretraining produces latent gains that only become visible after fine-tuning.

Their findings suggest that large-scale pretraining builds a core capability that fine-tuning unlocks for each downstream task. On a similar line of thinking, research on using reinforcement learning at test-time, for a purpose similar to TTFT here, indicates that it only reshapes the probability distribution of answers, as it only makes correct trajectories easier to hit (Yue et al. (2025)). It does not add any reasoning paths that were not there in the base model to begin with. The original GridCoder paper made a similar observation in the discussion section.

As for the gap between `TTFT+augments` and `TTFT-no-augments`, it is easily explained by the presence of geometrical data augmentations that cause the training procedure to see task examples that are very similar to the OOD task examples. The drop in performance for Task #7 is explained by the following: some training tasks consist of shifting pixels to the right, but none consist of shifting to the left. However, when the data augmentation horizontally flips or rotates by 180 degrees both the input and output grids to create a new task, what is presented effectively becomes the task of shifting pixels to the left. Thus, Task #7 is no longer out-of-distribution when this happens, explaining its high success rate.

Task #2 performance is explained by training task #13 (see Appendix C): it consists of shifting the pixels upwards by one row, and then vertically flipping the grid. Seen differently, this is the same as vertically flipping the grid first, and then shifting the pixels down by one row. Now, if one rotates the input and output grid by 270 degrees via data augmentation, a new task gets created: horizontally flip the grid, then shift the pixels to the right. This is exactly OOD Task #2.

In summary, the majority of the performance boost from using TTFT appears to come from its ability to elicit knowledge that comes from the training manifold, i.e. that is in-distribution with respect to it. We did not observe any substantial empirical evidence that TTFT is capable of learning novel tasks on-the-fly. These findings are coherent with the results from Teterwak et al. (2025), which indicate that transfer learning does not generalize to any unseen task. It improves performance when data distribution shifts are on the pretraining manifold; when shifts are orthogonal, fine-tuning is not sufficient.

### 7.2 SURPRISING SOLUTIONS

EG-NPS was able to derive unexpected solutions. In addition to being quite successful at discovering solutions to compositionally novel tasks (which was the expected and intended result), it also produced solutions that were surprising in two ways:

1. It composed subroutines in ways that omitted unnecessary instructions from one or both of the subroutines when the input grid's specific content allowed it.

2. It corrected redundancies in ground truth programs by ignoring the redundant instructions.

**Example #1**    In this example shown in Figure 2, the goal is to shift the pixels diagonally towards the upper-right corner of the grid. The given solution differs from the expected solution in two ways. First, unlike the ground truth "shift right" and "shift up" programs, no zeroing out of the leftmost column or the bottom row occur, because of the specificities of the input grid. Second, the solution did not contain a crop of the resulting grid after the "shift up" subroutine, contrary to the related ground truths.

Investigation revealed a bug in the ground truth programs, consisting of this redundant and superfluous `crop` operation. Although shifting right or down extends the grid and therefore requires cropping, shifting up or left silently discards pixels pasted to negative indices because of how the primitives have been implemented. Consequently, cropping is unnecessary in such cases.

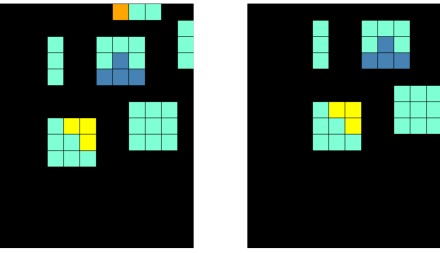

Figure 2: Example #1: the task consists of shifting the pixels diagonally towards the upper-right corner by 1 row, 1 column.

**Example #2**    This task (Figure 3) was implemented as a 180-degree rotation. Mathematically, composing a vertical and horizontal flip together produces the same result, which EG-NPS had no issue discovering. But the more interesting aspect is the efficiency of the solution. This is another case where it found a more efficient way to implement a routine than in our ground truth programs. It starts as expected by flipping the grid vertically. However, the subsequent steps implement a horizontal grid flip in a more condensed way than our ground truth program for this task, by doing it in 1 instruction instead of 3.

These 2 implementations are logically equivalent, but this is another case where our ground truth programs were inefficient. Indeed, our ground truth program was using an instruction `<colorOf>` that returns a list of pixel colors, instead of directly using the object attribute `<.c>` when setting the output grid pixels, which is more compact.

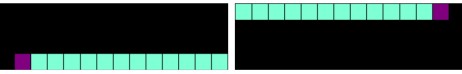

Figure 3: Example #2: the task consists of doing a 180-degree rotation. This can be also accomplished by flipping the grid both vertically and horizontally.

## 8    CONCLUSION

We conduct a controlled experiment that uses ARC-AGI-like tasks, evaluating the compositional generalization capabilities of various approches. Execution-guided NPS is the approach that was most able to extract compositional knowledge and to reuse it at test-time on novel tasks. In fact, it even innovated surprising solutions, with its ability to generate more efficient solutions than the program ground truths it was trained on.

In contrast, an ablation study that consists of omitting the test-time search component by only using the *argmax* solution significantly underperformed. GridCoder, which can be thought of as EG-NPS without the state-conditioned decoding, also significantly underperformed. Finally, experiments on TTFT identified no evidence of an ability to reliably generalize to structurally novel tasks.

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

## A APPENDIX A. SEARCH ALGORITHM PSEUDO-CODE

In the pseudocode, $SearchTreeNode$ is the class representing a node in the tree structure. The first argument is the state variable if we already have it, and the second argument is the parent node. The node's state variable can be set to None and later updated from inside the $execute\_program$ call. The latter executes the instruction step associated with the node, gets the resulting output, and updates the node's internal state variable. It also returns the result to verify if the goal was reached.

The $enumerate$ function aggregates encoder output from the parent node states and the target grid $Y$, and then uses the model $M$ to recursively query token probabilities and return possible token

---

**Algorithm 1** TreeSearch

---

**Input**: $X$, input grids
**Input**: $Y$, target grids
**Input**: $M$, the neural network model
**Parameter**: $\tau$, the time budget
**Parameter**: $\epsilon$, the maximum tree depth
**Output**: The solution, if found.

```
 1: // Instantiate the root node
 2: root = selected = SearchTreeNode(X, None)
 3: start_time = time()
 4: while time() − start_time < τ do
 5:    output = execute_program(selected)
 6:    if output == Y then
 7:       // Solution found!
 8:       return selected
 9:    end if
10:    progs = enumerate(M, Y, selected)
11:    for all prog ∈ progs do
12:       log_prob = calculate_joint_prob(prog)
13:       Insert into program queue: (prob, prog)
14:    end for
15:    next_best = Dequeue program queue
16:    leaf = program leaf node for next_best
17:    child = SearchTreeNode(None, leaf)
18:    leaf.child = child
19:    selected = child
20: end while
21: return None
```

---

sequences for the next instruction step whose probability is greater than some threshold. We use 0.01 in our experiments. Also, if the number of generated possible sequences exceeds a threshold $k$, we only keep the top $k$ (k = 1M in our experiments). Finally, there is a maximum token sequence length of 40 tokens.

Suppose *enumerate* generates 42 different possible instruction steps at this stage in the program, this means that we add 42 new programs to the program queue: each starting with the same instruction steps for the node parents trajectory, and ending with the newly generated instruction steps.

Not presented in the pseudo-code (for simplicity) are further search constraints such as maximum tree depth (20 in our experiments) and a memory limit of 5. This memory limit refers to the maximum number of intermediate state variables that can be kept in the program's working memory at all times. When that memory limit is reached, only the `del` primitive is allowed to clear up some memory.

## B    APPENDIX B. HYPERPARAMETERS

**Execution-guided NPS**    The neural network used in the experiments presented in this paper is a standard Encoder-Decoder transformer architecture (using PyTorch), with 4 encoder and 4 decoder layers. The feedforward layer's dimensionality is 1024, the embedding space (d_model) has a dimension of 256, and 16 attention heads were used. The input vocabulary has a size of 55 and the target vocabulary has a size of 45.

**TTFT**    Fine-tuning from scratch on our training data was not done using LoRA. Instead, it was a full, "standard" training of the weights. While most of the hyperparameters used were the same ones used by Omni-ARC to reach 2nd place in the 2024 ARC-AGI competition, in an attempt to improve the performance of `TTFT-no-augments` we varied some hyperparameters in the test-time fine-tuning phase. In particular:

1. max_steps: we tried fine-tuning a different number of steps from 10 steps up to the maximum number of steps that took 2 minutes 30 seconds, to respect the 3-minute time budget (the subsequent LoRA merge step, inference step and voting steps took approximately 30 seconds). This lead to trying up to about 60 fine-tuning steps.

2. learning rate: varied between `2e-4` to `1e-5` in various increments.

3. batch_size: we tried the default 16, 8 and 32.

4. eval_steps: we tried 10, 20, 50.

## C  APPENDIX C. OOD TASK DETAILS

The neural network is trained on generated training data that implements the following 14 simple ARC-AGI-like tasks, on grids of variable dimensions between $3 \times 3$ and $30 \times 30$:

1. Horizontally flip the grid around the vertical axis formed by the center column.

2. Vertically flip the grid around the horizontal axis formed by the center row.

3. Set all non-zero, non-black pixels to the color green.

4. Horizontally flip the grid and set all its foreground pixels to green.

5. Vertically flip the grid and set all its foreground pixels to green.

6. Shift pixels to the right by 1 column, with no wrapping; the leftmost column is left completely black.

7. Shift pixels upward by 1 row (no wrapping).

8. Shift pixels downward by 1 row (no wrapping).

9. Shift pixels to the right by 1 column, and then horizontally flip the resulting grid around the central axis.

10. Vertically flip the grid, and then shift the resulting grid to the right by 1 column.

11. Horizontally flip the grid, and then shift the pixels upward by 1 row.

12. Shift the pixels downward by 1 row, and horizontally flip the grid.

13. Shift the pixels upward by 1 row, and then vertically flip the grid.

14. Vertically flip the grid, and then shift the pixels upward by 1 row.

The OOD tasks are:

1. Shift the pixels to the right by 1 column, and set all the non-zero pixels to green.

2. Horizontally flip the grid, and then shift the pixels to the right by 1 column.

3. Shift the pixels diagonally towards the upper-right corner by 1 cell.

4. Rotate 180 degrees

5. Horizontally flip the grid, shift it to the right, then vertically flip it.

6. Set the non-zero pixels to green, shift diagonally the pixels towards the upper-right corner.

7. Shift pixels to the left (no wrapping).

## D  APPENDIX D. DSL DETAILS

Object attributes:

1. .x: a list of all x coordinates of each grid cell from left to right, top down

2. .y: a list of all y coordinates of the grid from left to right, top down

3. .c: a list of all pixel colors of the grid from left to right, top down

4. .width: the number of columns in the grid

5. .height: the number of rows in the grid

6. .max_x: width - 1 (essentially a shortcut to simplify code)

7. .max_y: height - 1

8. .ul_x: x coordinate of the upper left corner of this grid. Can be non-zero if this is a sub-grid representing an object in the outer grid.

9. .ul_y: y coordinate of the upper left corner of this grid. Can be non-zero if this is a sub-grid representing an object in the outer grid.

# E    APPENDIX E. LLM USAGE IN WRITING THIS PAPER

ChatGPT has been used in reformulating and polishing sections of this text that were deemed a bit "rough" or complicated. No "new content" was added by ChatGPT. In particular, the sections that where polished using an LLM were:

1. The abstract

2. The first two paragraphs of the introduction

3. The related work section

4. The tables of the DSL section

5. The conclusion

