# OpenReview forum: "Out-of-Distribution Generalization in the ARC-AGI Domain: Comparing Execution-Guided Neural Program Synthesis and Test-Time Fine-Tuning"
_ICLR.cc/2026/Conference — Submitted to ICLR 2026_

### Official Review · Reviewer_dP2F · 2025-10-20

**Soundness:** 3
**Presentation:** 3
**Contribution:** 2
**Rating:** 6
**Confidence:** 4

**Summary:**

The authors propose to study the ability of execution-guided program synthesis approach and transduction approaches with test-time training to generalize to new ARC-AGI-like tasks at test time. Train and test tasks are designed by hand to involve different compositions of the same set of predefined primitives.

Overall, the paper asks an interesting and timely question about compositional generalization in ARC-like domains and provides a well-controlled experimental setup. The execution-guided synthesis results are solid and the TTFT analysis is carefully done. However, the scope of the comparison is narrow: both methods rely on hand-crafted DSLs and synthetic data, the OOD tasks are simple, and the advantage of EG-NPS may mostly reflect its explicit search and verification loop rather than deeper generalization. The study is informative within its controlled sandbox but limited in how much it says about solving the broader ARC-AGI challenge or about generalization in pretrained LLMs.

**Strengths:**

I liked that the paper carefully controlled their experiments by designing all tasks by hand, making sure test tasks involved new combinations of primitives that were not encountered at training time.

The test-time training experiment is particularly well designed, controlling for different possible factors explaining its success in prior work.

This kind of question is interesting and deserves more investigation, and ARC-AGI might be an interesting domain to conduct such experiments.

**Weaknesses:**

**Missing related work:**

The related work section (and intro) miss significant related work:
* CodeIt: Self-Improving Language Models with Prioritized Hindsight Replay: https://arxiv.org/abs/2402.04858v2
* Self-Improving Language Models for Evolutionary Program Synthesis: A Case Study on ARC-AGI: https://arxiv.org/pdf/2507.14172
* Combining Induction and Transduction for Abstract Reasoning: https://arxiv.org/abs/2411.02272

“Execution-guided, neural program synthesis has not yet been attempted”: The second paper above applies a genetic algorithm to solve ARC, with mutation operators conditioned on previous solutions and their execution feedback. In general, any LLM-based GA method (FunSearch, AlphaEvolve, etc.) is “execution-guided”. The authors seem to use a different meaning — more about search within program execution rather than post-execution feedback. Did I get this right? This should be clarified.



**Test-time training and program synthesis are orthogonal**
The third paper above already separates program synthesis (induction) from direct generation (transduction). This axis is orthogonal to test-time training. One can have a program-synthesis model that also learns at test time (as in the last two papers above), or a transductive model without any TTT. So opposing “program synthesis” and “TTT” doesn’t seem to be the key conceptual split here.
In the discussion of Akyürek et al., it says “they generate tasks synthetically … that can overlap with validation data.” I think that’s fine — if a system can imagine validation-like data and improve from it, that’s still generalization. It only becomes a problem if augmentations are designed knowing the validation set.

**Solving ARC-AGI ≠ recombining known primitives**
This paper studies recombination, but ARC-AGI itself doesn’t come with a DSL. True generalization may require inventing new operations at test time, not just reusing old ones. The DSL here can only solve ~18 ARC-AGI-2 tasks, so conclusions about full ARC-AGI generalization are limited. Moreover, the “OOD” tasks (flips, shifts, recolors) cover a narrow subset of ARC reasoning.

**Dependence on hand-crafted DSL and synthetic data**
Both methods depend on a fixed DSL and synthetic training data. The DSL defines what “generalization” means: no new primitives can be invented. The training distribution (types of transformations, grid sizes, augmentations) also determines what counts as OOD. No ablation checks sensitivity to DSL or data design. Code and primitive lists are missing, making reproduction complicated:
* What are the DSLs with 34 and 45 primitives?
* What are the training tasks?
* What are the training tasks and primitives which were added for the ARC-AGI-2 experiment, and why?

**TTT interpretation**
The claim that TTFT only surfaces pre-existing abilities is more of an interpretation rather than demonstrated. It may simply be that pre-training + TTT together enable generalization. Without knowing the pre-training corpus of course, this can’t be verified.
For the same reason, the study can’t say much about LLM-based program-synthesis methods that rely on large pretrained models (like the ones developed in the two last papers of the above list): these would be hard to train from scratch under a controlled setup.

**Other points**
The claim that humans solve these tasks should be supported by data (e.g. http://arxiv.org/pdf/2409.01374). Can any human solve them, or at least some? I think in ARC-AGI 2 it’s more of “out of a group of N, at least 2 can solve the task”.

**Questions:**

* What exactly is meant by “open-world” in this context? ARC-AGI seems quite closed in scope.
* What are the different DSL and training tasks?
* Could the observed advantage of EG-NPS be mostly due to its explicit search and verification loop, which the TTT baselines can’t really do?

---

> ### Author Response · Authors · 2025-11-12
> **Responses to questions and comments**
>
> Thank you for the detailed and in-depth review. We appreciate the thought-provoking comments and questions.
>
> Regarding the missing related work: Thank you for this information. We will read them and see if it makes sense to include them in the Related Work section. We were already aware of the "Combining Induction and Transduction for Abstract Reasoning" and felt the content was orthogonal to this paper, but we will reconsider based on your suggestion.
>
>     “Execution-guided, neural program synthesis has not yet been attempted”: The second paper above applies a genetic algorithm to solve ARC, with mutation operators conditioned on previous solutions and their execution feedback. In general, any LLM-based GA method (FunSearch, AlphaEvolve, etc.) is “execution-guided”. The authors seem to use a different meaning — more about search within program execution rather than post-execution feedback. Did I get this right? This should be clarified.
>
> Your understanding is correct. We are aware of existing approaches that run the program, get feedback, make modifications to the program, and re-try, etc. (e.g. Greenblatt, which we cite). The proposed EG-NPS has a fundamentally different mechanism, as you pointed out. We believe the term "execution-guided" has a narrower meaning, as in the cited Related work literature (starting with Zohar & Wolf (2018)) -- where you are testing the effect of individual instructions as you build a program candidate.
>
>     Test-time training and program synthesis are orthogonal. The third paper above already separates program synthesis (induction) from direct generation (transduction). This axis is orthogonal to test-time training.
>
> This point is well received. We did not intend to create a false dichotomy between program synthesis and TTFT like that. The choice of TTFT as the main "opponent" here is based mostly on the immense popularity of this approach at the time of writing the paper and the authors' suspicion that it did not generalize OOD as much as it was claimed. We did not intend to suggest that these two approaches are mutually exclusive. We will review and correct any wording that explicitly suggests this (if we did at any point in the paper).
>
>     Solving ARC-AGI ≠ recombining known primitives. This paper studies recombination, but ARC-AGI itself doesn’t come with a DSL. True generalization may require inventing new operations at test time, not just reusing old ones. (...) Dependence on hand-crafted DSL and synthetic data Both methods depend on a fixed DSL and synthetic training data. The DSL defines what “generalization” means: no new primitives can be invented.
>
> This is an opinion, or philosophical perspective, that we disagree with. We adhere to the Language of Thought hypothesis according to which there are fundamental axioms or building blocks of reasoning -- and all reasoning is a composition of these. As an analogy, the entire complexity of the universe derives from just a "few" fundamental rules, axioms and principles. No new fundamental principles ever get created -- only novel compositions thereof. So we find this idea of reasoning as compositional generalization from fundamental building blocks so be intuitively plausible and compelling.
>
>     Code and primitive lists are missing, making reproduction complicated. (...) What are the training tasks? (...) What are the different DSL and training tasks?
>
> Please see the supplementary material attached to this submission, it is all there. The source code is included along with the DSL. The pdf explains conceptually what are the training tasks and the OOD tasks.
>
>     What exactly is meant by “open-world” in this context? ARC-AGI seems quite closed in scope.
>
> open world refers to a problem domain where the full scope of possible states cannot be covered in practice. As a result, it's not possible to train a neural network to interpolate the full space of solutions in that world. Or, to use the definition provided by ChatGPT: "The model is trained on a known, finite set of classes or conditions, but during deployment, it may encounter samples from unknown or novel classes, distributions, or contexts, and the system must detect, adapt to, or learn these new inputs without assuming the world is closed." We believe ARC-AGI fits well into that category.
>
>     Could the observed advantage of EG-NPS be mostly due to its explicit search and verification loop, which the TTT baselines can’t really do?
>
> We think the paper demonstrates that the advantage of EG-NPS is that it can generate solutions that are structurally novel compared to what it has seen during training. Otherwise, it could not solve the OOD tasks. This ability is enabled by the fact that as it initially attempts arbitrary solutions on the OOD task, and eventually it often stumbles on an intermediate state that it then knows how to carry over to the target state.

---

> > ### Comment · Reviewer_dP2F · 2025-11-24
> >
> > Thank you for this detailed response
> >
> > I did miss the supplementary zip, thank you for pointing it out. The definition of execution-guided is now clarified.
> >
> > I'm not sure this definition of open-world is very standard and it should maybe be explicited in the paper. Here it is used to argue that neural nets (TTT) can't generalize to OOD test tasks because, in open-worlds, test tasks are expected to be "from unknown or novel classes, distributions, or contexts,  and the system must detect, adapt to, or learn these new inputs." I believe this kind of situation is also going to be a problem for the alternative approach proposed here. As the approach relies on a fixed DSL, the OOD test tasks might rely require operations not present in the DSL that would make the approach strictly incapable of solving the OOD task.
> >
> > The language of thought perspective assumes the existence of a universal DSL that would allow all possible transformations, and such a DSL probably exists, but is it searchable? Is it really OOD generalization if you hand-code a DSL that you know is able to capture OOD task transformations? It is OOD in the sense of "new recombinations of known primitives" but is not in the sense of "new primitives altogether."
> >
> > ARC was specifically designed without a DSL, to make it more complicated to simply reverse engineer a perfect DSL to solve all ARC tasks. It might still be possible to find one of course, but so far it seems no one has.
> >
> > Overall I think these two points, part of the central claims, are not really demonstrated in the paper:
> > * The proposed approach can generalize to strong OOD tasks (meaning involving unknown primitives, not just new combinations)
> > * The TTT approach cannot generalize to OOD tasks (as argued in my first message, this paper only shows that TTT trained from scratch have a hard time generalizing to OOD tasks
> >
> > Because the DSL is hand-designed by the experimenters, who also designed the OOD test tasks, it feels like this "weak" form of generalization (recombination of known primitives) is kind of baked in the DSL+search
> >
> > This said I think this is a step in the right direction and it's already interesting to see how search might perform better OOD than TTT trained from scratch.
> > But I believe that these claims would only be supported by scaling the execution-guided search approach: either providing a DSL able to let the approach generalize to real ARC-test tasks ("strong OOD"), or adding a DSL construction mechanism

---

> > > ### Author Response · Authors · 2025-11-24
> > > **Response to the critique of DSL-based approaches**
> > >
> > > I'm not sure this definition of open-world is very standard and it should maybe be explicited in the paper. Here it is used to   argue that neural nets (TTT) can't generalize to OOD test tasks because, in open-worlds, test tasks are expected to be "from unknown or novel classes, distributions, or contexts, and the system must detect, adapt to, or learn these new inputs." I believe this kind of situation is also going to be a problem for the alternative approach proposed here. As the approach relies on a fixed DSL, the OOD test tasks might rely require operations not present in the DSL that would make the approach strictly incapable of solving the OOD task.
> > >
> > > This is best understood when considering two degrees of OOD generalization. The weaker form of OOD generalization is that the target task leverages the same fundamental primitives, but composes them in a way that is novel (compositional generalization). In the stronger form of OOD generalization, even the primitives are new. This paper addresses the first kind, and we agree that even our approach does not generalize to the second kind. The main argument in our paper is that our proposed approach is demonstrably better than a pure NN approach (even with TTFT) at generalization to the weaker OOD case.
> > >
> > > Also, it's not clear to us that anything can solve the stronger form of OOD (not even humans) -- although that debate probably gets philosophical very quickly.
> > >
> > >     The language of thought perspective assumes the existence of a universal DSL that would allow all possible transformations, and such a DSL probably exists, but is it searchable? Is it really OOD generalization if you hand-code a DSL that you know is able to capture OOD task transformations? It is OOD in the sense of "new recombinations of known primitives" but is not in the sense of "new primitives altogether."
> > >
> > > As an example, Python is Turing complete and contains all the necessary primitives to implement all the algorithms we need to implement, using only 36 keywords and 44 operators. From this very finite set of primitives we can compositionally generate all algorithms we can think of. Thus it is safe to expect that there are finite "ARC-AGI complete" languages (of which Python is such a language, because ARC-AGI is a subset of Turing complete, it's just not a very efficient one for the ARC-AGI problem). Michael Hodel has already an ARC-AGI DSL which is arguably very complete (though it's not proven it can implement all possible tasks). All of this makes a strong case that a finite set of primitives is needed, after that it's all compositional generalization from there.
> > >
> > >     ARC was specifically designed without a DSL, to make it more complicated to simply reverse engineer a perfect DSL to solve all ARC tasks. It might still be possible to find one of course, but so far it seems no one has.
> > >
> > > While it's true that no DSL has been provided for ARC, its creator (François Chollet) is a strong proponent of program synthesis, and a major skeptic of pure NN-based approaches. Therefore I doubt that the intent of ARC-AGI was specifically to make it more complicated to reverse engineer a DSL from it. Instead the main motivation was to show that pure interpolation/learning-based approaches are not sufficient.
> > >
> > > For the second part of this comment: Python is a DSL that can solve all ARC tasks (though not in a very efficient way). Michael Hodel's DSL has very good coverage (all ARC-AGI v1 training tasks, and possibly evaluation tasks, have been implemented using his DSL).
> > >
> > > Let's put this differently: if we had chosen Python as the programming language, this line of critique breaks down. It is not particularly controversial to say that Python can solve all, or at the very least most, of ARC-AGI tasks. Once this is accepted, what is wrong with trying to create a more efficient programming language for ARC-AGI? Or is the critique that we are using a custom language instead of Python?
> > >
> > >     Overall I think these two points, part of the central claims, are not really demonstrated in the paper:
> > >     - The proposed approach can generalize to strong OOD tasks (meaning involving unknown primitives, not just new combinations)
> > >
> > > Agreed, it it also not our intent to claim that it can generalize to the strong case of OOD tasks. However, the weaker case of OOD tasks is potentially accessible, and it is a blocking point for most approaches, so it is an interesting research problem. We demonstrate some success in that area.
> > >
> > > Critique of the TTT experiments will be addressed separately

---

> ### Author Response · Authors · 2025-11-24
> **Response to the critique of the TTT experiments/interpretation**
>
> Regarding the comment: "Overall I think these two points, part of the central claims, are not really demonstrated in the paper: (...)
>      - The TTT approach cannot generalize to OOD tasks (as argued in my first message, this paper only shows that TTT trained from scratch have a hard time generalizing to OOD tasks)"
>
> We probably agree that if the TTT model has been pretrained on similar tasks, its ability to solve these tasks at test-time is not interesting -- so we interpret this criticism as implying that you believe a set of foundational, implicit primitives could be learned from pretraining on an extremely large dataset (even if it has never seen ARC-AGI tasks, for example), and then at test time, TTT could compositionally discover the solutions in the same way that our approach could. Is that the correct implication? Furthermore, we believe you hinted at the fact that maybe this is what is happening in the fully pretrained case in our experiments?
>
> If so, we definitely cannot confirm that this is impossible, and we generally use careful language to that effect in the paper:
>
> "Finally, experiments on TTFT identified no evidence of an ability to reliably generalize to structurally novel tasks." (instead of saying that we have evidence that it cannot)
>
> "It is conceivable that it has already been trained on tasks that are sufficiently similar to relatively simple operations found in our OOD tasks" (instead of saying that we know for a fact that the LLM was pretrained on these tasks)
>
> "We did not observe any substantial empirical evidence that TTFT is capable of learning novel tasks on-the-fly"
>
> etc.
>
> One cannot empirically prove a negative: so we could never prove that TTFT cannot generalize OOD. In the eyes of the believers it will always be "not enough data", "model not big enough", "more data diversity", "wrong model", etc. The best we can do, scientifically, is to see if we can reject the null hypothesis that TTFT does not help a base model to generalize OOD (thereby proving that TTFT helps generalize OOD). In this case, we could not reject it. It would be interesting to see if other people can replicate this experiment, or maybe find a way to reject the null hypothesis in a methodologically rigorous way (which we have not yet seen).

---

### Official Review · Reviewer_RKf8 · 2025-10-31

**Soundness:** 3
**Presentation:** 3
**Contribution:** 2
**Rating:** 4
**Confidence:** 3

**Summary:**

This paper investigates out-of-distribution (OOD) generalization on the ARC-AGI benchmark by comparing execution-guided neural program synthesis (EG-NPS) with test-time fine-tuning (TTFT). The authors introduce a novel EG-NPS algorithm alongside a custom domain-specific language (DSL). Through controlled experiments, EG-NPS significantly outperforms non-execution-guided methods and TTFT on a series of experimental setups. The results demonstrate that execution-guided synthesis enables stronger compositional reasoning and more robust OOD generalization than fine-tuning-based approaches in the ARC-AGI domain.

**Strengths:**

The paper is overall well-written:
- Focus on compositional and OOD generalization.
- Novel tree search algorithm for execution-guided neural program synthesis.
- Diverse empirical setups: include baselines and ablations.

**Weaknesses:**

The paper’s main weaknesses include scale, scope, and generalizability of the experiments:
- Requires ground-truth programs as training data.
- Requires a DSL, and the implemented one is very limiting.
- Limited scale and empirical scope: for instance, experiments on ARC-AGI include only 18 tasks out of the 1000+ available.
- Other fully neural baselines could have been tested against: e.g., [Ouellette, 2024] or [Bonnet & Macfarlane, 2024].

**Questions:**

How would you think of scaling the proposed approach? Suggesting a few directions for future work could improve the paper.

---

> ### Author Response · Authors · 2025-11-12
> **Responses to questions and comments**
>
> Regarding weaknesses:
>
> The need for ground-truth programs and a DSL are indeed weaknesses of program synthesis approaches. That being said, it is a valid research avenue, as it has counter-balancing strengths compared to purely neural approaches. And indeed, in this paper, we showcase some of those advantages.
>
> We agree and recognize that the main limitation of our paper is the limited experimental scope. We prioritize spending our time and effort on improving our approach over widening our experimental scope, and felt that the current results could be sufficiently interesting "as is" to warrant being published.
>
> Regarding the question:
>
> The main blocking points to scaling that we identified are:
> 1. making the DSL sufficiently expressive to cover all (or the vast majority) of possible tasks. The biggest blocking point specifically is the ability to recognize/identify objects, which we did not have in this paper. We estimate that we can reach that full DSL in less than 100 primitives.
> 2. getting training data is more effortful than transductive approaches that only need the input grids and output grids: we also need the program ground truths written in our custom DSL. We are currently researching "self-play" or "semi-self-supervised" methods to partially automate ground truth generation.
>
> We will see if we can squeeze this information into an updated version of the paper (since we're already hitting the page limit).

---

> > ### Comment · Reviewer_RKf8 · 2025-11-15
> >
> > Thank you for your answers.
> >
> > For completeness, not all program synthesis approaches suffer from this DSL requirement limitation. For instance, LPN [Macfarlane & Bonnet, 2025], HRM [Wang et al., 2025], and TRM [Jolicoeur-Martineau, 2025] do not rely on a DSL to work on ARC-AGI or at least do not rely on program ground truths.
> >
> > Thank you for addressing the question. Another point regarding the dependence on DSL is that solving complex tasks may require the invention of primitives, not just recombination. The manuscript would be significantly improved by detailing and tackling some of these limitations.
> >
> > - Matthew V Macfarlane, Clement Bonnet. Searching Latent Program Spaces. 2025.
> > - Alexia Jolicoeur-Martineau. Less is More: Recursive Reasoning with Tiny Networks. 2025.
> > - Wang, G., Li, J., Sun, Y., Chen, X., Liu, C., Wu, Y., Lu, M., Song, S., and Yadkori, Y. A. Hierarchical reasoning model. 2025.

---

> > > ### Author Response · Authors · 2025-11-16
> > > **Regarding the suggested literature**
> > >
> > > We already cite Matthew V Macfarlane, Clement Bonnet. Searching Latent Program Spaces. 2025 in our Related Work section. We are aware of the TRN and HRM papers but did not think of them as program synthesis per se, more like "reasoning" models. As far as we are aware, the TRN paper does not frame their approach as a kind of program synthesis, but the analogy is indeed interesting. We will give these approaches some consideration as potential future improvements.
> > >
> > > To be clear, we see the automatic learning of primitives are the ultimate goal in our research. The use of an explicit hand-crafted DSL is currently an intermediate stepping stone for us, while we focus on the ability to efficiently search for out-of-distribution solutions. Having an explicit DSL has advantages as far as troubleshooting and understanding what is happening "under the hood" in a program synthesis solution, especially when analyzing the generalization capabilities of different approaches.

---

### Official Review · Reviewer_2M9h · 2025-11-01

**Soundness:** 3
**Presentation:** 4
**Contribution:** 2
**Rating:** 4
**Confidence:** 3

**Summary:**

The paper target the problem of solving out-of-distribution (OOD) tasks in the ARC-AGI-2 visual reasoning benchmark. The approach extends the line of work on execution-guided program synthesis with an approach the paper calls EG-NPS. It uses a simple DSL which integrates with a standard encoder-decoder transformer as the program generator. The idea is to generate instructions and to partially evaluate them, exposing both of these as execution trace to program generator. This informs the search procedure, which is a tree search with tunable randomness, to complete the program that reaches the end goal. The system is trained on ground truth programs that successfully solve some task. The evaluation is on different tasks in the same visual domain.

The experiments compare with other competing approaches which do not carry all the features of EG-NPS. The results show significant gains on 7 task that were evaluated during tests.

**Strengths:**

* The paper is well written and explains enough details to understand the distinction to prior approaches.

* The ablation study is focused and explains where EG-NPS improves over prior works evaluated.

* The gains on the programs evaluated in success rate are significant. The approach solves about 83% of the few (1.5%) of ARC-AGI-2 tasks within scope of the current DSL considered in the work.

The paper's proposed methods combines a number of pre-existing ideas, but in a systematic way. The analysis and design of experiments is performed carefully to understand where the gains are coming from.

**Weaknesses:**

* The paper is relatively weak on explaining how the expressiveness of the DSL is a stumbling block to a broader evaluation. The paper briefly mentions in Section 6.4 that the DSL had to expanded to deal with just 1.5% of ARC-AGI-2 benchmarks. It is known that the search space explodes and there is work characterizing the input-output samples needed therein. Please see the work by Wang et al. [a]. It will be useful if the paper talks about the issue of DSL design, state explosion, and expected sample complexity.

* The paper is scoped to a particular benchmark. While this is a completely reasonable way to devise a better program synthesis technique, the resulting ideas are in no way restricted to visual tasks. Clearly the tree-search method, transformer for synthesis, equipped with a DSL can be targeted to other real-world tasks. Did you consider evaluating your proposed methods on any other non-ARC-AGI-2 task? For example, you could take a standard DSL (e.g. FlashFill), architectural instruction set, or any other minimal programming language and try to solve coding tasks. This kind of experiments reduce the risk of ideas being overfit to a benchmark or set of tasks.

* The paper references Lavon et al. (2025). This is execution-guided technique too. I understand that EG-NPS is targeting ARC-AGI-2, but at the technical level both are execution-guided techniques. As such, it would be useful to see a comparison on common benchmarks. To understand the advance presented by EG-NPS, it is useful to run it on Lavon et al.'s evaluated benchmarks.

[a] "SynGuar: Guaranteeing Generalization in Programming by Example", Wang et al., In ESEC/FSE 2021: Proceedings of the 29th ACM Joint Meeting on European Software Engineering Conference and Symposium on the Foundations of Software Engineering.

**Questions:**

Questions are raised in the weakness section.

---

> ### Author Response · Authors · 2025-11-12
> **Responses to comments and questions**
>
> Regarding the first part of the first weakness comment, the main blocking point is that our DSL did not have a primitive to extract objects. In other words, it had no notion of objectness, but most ARC-AGI tasks are transformations/operations on objects in a grid, rather than the grid itself as a set of pixels. This greatly limits our coverage.
>
> Regarding the second part of the first weakness comment, it is indeed interesting to consider the theoretical aspects of DSL design and search space explosion, but we have not done that work yet. We will read the provided reference, thank you for the information.
>
> Regarding being scoped to only 1 benchmark (ARC-AGI), we fully agree. We recognize that the biggest limitation of our paper (in our opinion) is that we did not add different benchmarks outside of ARC-AGI. We were hoping the findings to be interesting enough "as is", but of course that is not for us to decide.

---

### Official Review · Reviewer_ocig · 2025-11-05

**Soundness:** 3
**Presentation:** 3
**Contribution:** 2
**Rating:** 6
**Confidence:** 3

**Summary:**

The paper considers compositional generalization within the the ARC-AGI domain. The authors demonstrate that execution-guided program synthesis performs best, beating out other neural program synthesis approaches as well as test-time fine-tuning of LLMs. To achieve this, the authors developed a DSL for the ARC-AGI domain.

**Strengths:**

1. Compositional generalization is an important yet underexplored problem. The setup the authors considered in ARC-AGI will be generally useful for evaluating compositional generality of future approaches.

2. The authors demonstrate that fine-tuning is primarily useful for unlocking latent in-distribution knowledge and not OOD generalization. This is an interesting finding as such fine-tuning approaches are very popular in practice.

3. The authors develop (to my knowledge) the first instance of execution-guided neural program synthesis for the ARC-AGI domain with its own DSL. This will be a useful baseline to compare against for this benchmark.

**Weaknesses:**

1. While execution-guided program synthesis is proven to be effective at compositional generalization, its applicability seems to be limited due to the requirement of designing a DSL that covers all theoretically solvable tasks. In particular, the DSL the authors designed does not cover the full scope of ARG-AGI tasks. So it is unclear how useful such approaches will be in general.

2. The entropy term added to search did not seem particularly effective. It is unclear if the search algorithm implemented was suboptimal.

3. Furthermore, my understanding is that neural program synthesis and fine-tuning consider very different test-time search algorithms. This makes the results a bit more difficult to parse, as it is unclear how important the choice of search algorithm was.

**Questions:**

1. Have the authors considered training neural program synthesis from a pretrained LLM rather than from scratch? While one would expect it to not make much of a difference, it would be nice to disentangle the base architecture from the training algorithm.

2. If I understand correctly, the other neural program synthesis approach GridCoder uses a different DSL than EG-NPS. What would happen if the DSL used between the two approaches were the same?

---

> ### Author Response · Authors · 2025-11-12
> **Response to comments and questions**
>
> Regarding weaknesses:
>
> 1. We could argue this is a weakness of DSL-based program synthesis in general, rather than specifically this paper. It is true that the requirement for a DSL is prohibitive in the amount of engineering effort that is involved. However this approach is, for us, a stepping stone. In the long-run we hope to find a way to learn these primitives automatically.
>
> 2. The entropy functionality is not an important or "core" component of our proposed approach. It was an ad hoc addition once we realized that sometimes the algorithm covers the full space of "likely programs" according to the NN in under the theoretically allocated time of 3 minutes according to the ARC-AGI competition. We do not claim that this particular functionality is optimal, or even interesting. Our paper stands on its own without this functionality.
>
> 3. We're not sure to understand this comment, but the main goal was to show that TTFT is not conducive to genuine discovery of new tasks, while the proposed form of program synthesis is. We believe this was shown in quite an in-depth manner, for example with the decomposed experiments on TTFT that show that most of the performance gains come from the prior knowledge of the LLM itself (and some of it due to the hand-crafted data augmentations). In contrast, we also clearly showed that the EG-NPS approach proposed is able to go beyond its training distribution and find solutions to structurally novel tasks. This is the crux of our paper.
>
> Regarding questions:
>
> 1. This was not done in the scope of this paper, no. However, since then we have tried this and, as you rightly suspect, it did not make much of a difference. We found it difficult to leverage the prior knowledge built into the pretrained LLM via program synthesis (possibly because of the way that the LLM was pretrained, maybe with an emphasis on transduction "grid-to-grid" prediction rather than program synthesis).
>
> 2. Correct. The GridCoder uses a much higher level DSL, meaning that it is less flexible but makes for shorter programs. We found this DSL to be too specific, and aimed for a much lower-level, more general and granular DSL. This was actually a benchmark handicap for our approach, because for the tasks we experimented on the search space is much larger in our DSL than in the GridCoder DSL. In other words: all the tasks we experimented on have solutions in both DSLs. However, because the GridCoder DSL has more hand-crafting work built-in (more complex primitives) it can represent solutions in less instructions. Yet, in spite of that, our approach largely out-competed GridCoder.

---

### Meta-Review · Area_Chair_PKA4 · 2026-01-12

**Summary:**

The DSL approach has limited scope.

**Reviewer Concerns:**

The most important concern is that the DSL approach has limited scope.

**Reviewer Scores:**

I can't tell on behalf of the reviewers, since half of them didn't reply to the rebuttal.
The ones who did, didn't seem convinced.

---

### Decision · Program_Chairs · 2026-01-26

Reject